# Nuclear corepressor SMRT acts as a strong regulator of both β-oxidation and suppressor of fibrosis in the differentiation process of mouse skeletal muscle cells

Hiroaki Shimizu[1]*, Yasuhiro Horibata[1], Izuki Amano[2,3], Megan J. Ritter[3], Mariko Domae[4], Hiromi Ando[1], Hiroyuki Sugimoto[1], Ronald N. Cohen[5], Anthony N. Hollenberg[3]

1 Department of Biochemistry, Dokkyo Medical University School of Medicine, Shimotsuga-gun, Tochigi, Japan, 2 Department of Integrative Physiology, Graduate School of Medicine, Gunma University, Showa, Maebashi, Gunma, Japan, 3 Department of Medicine, Weill Cornell Medical College, New York, NY, United States of America, 4 Center for Research Collaboration and Support, Comprehensive Research Facilities for Advanced Medical Science, Dokkyo Medical University School of Medicine, Shimotsuga-gun, Tochigi, Japan, 5 Department of Adult and Pediatric Endocrinology, Diabetes and Metabolism, The University of Chicago Medicine, Chicago, IL, United States of America

* shimizuh@dokkyomed.ac.jp

**Data Availability Statement:** All relevant data are within the manuscript and its Supporting Information files.

## Abstract

### Background

Silencing Mediator of Retinoid and Thyroid hormone receptors (SMRT; NCoR2) is a transcriptional corepressor (CoR) which has been recognized as an important player in the regulation of hepatic lipogenesis and in somatic development in mouse embryo. SMRT protein is also widely expressed in mouse connective tissues, for example adipocytes and muscle. We recently reported that mice with global deletion of SMRT develop significant obesity and muscle wasting which are independent from thyroid hormone (TH) signaling and thermogenesis. However, the tissue specific role of SMRT in skeletal muscle is still not clear.

### Methods

To clarify role of SMRT in muscle differentiation, we made myogenic C2C12 clones which lack SMRT protein (C2C12-SKO) by using CRISPR-Cas9. Wild-type C2C12 (C2C12-WT) and C2C12-SKO cells were cultured in differentiation medium, and the resulting gene and protein profiles were compared between the two cell lines both before and after differentiation. We also analyzed muscle tissues which were dissected from whole body SMRT knockout (KO) mice and their controls.

### Results

We found significant up-regulation of muscle specific β-oxidation markers; Peroxisome proliferator-activated receptor δ (PPARδ) and PPARγ coactivator-1α (PGC-1α) in the C2C12-SKO cells, suggesting that the cells had a similar gene profile to what is found in exercised

**Funding:** Financial Support: This work was supported by JSPS KAKENHI Grant number; 18K08486 and 21K07347. The funders had no role in study design, data collection and analysis, decision to publish, or preparation of the manuscript.

rodent skeletal muscle. On the other hand, confocal microscopic analysis showed the significant loss of myotubes in C2C12-SKO cells similar to the morphology found in immature myoblasts. Proteomics analysis also confirmed that the C2C12-SKO cells had higher expression of markers of fibrosis (ex. Collagen1A1; COL1A1 and Fibroblast growth factor-2; FGF-2), indicating the up-regulation of Transforming growth factor-β (TGF-β) receptor signaling. Consistent with this, treatment with a specific TGF-β receptor inhibitor ameliorated both the defects in myotube differentiation and fibrosis.

## Conclusion

Taken together, we demonstrate that SMRT functions as a pivotal transcriptional mediator for both β-oxidation and the prevention for the fibrosis via TGF-β receptor signaling in the differentiation of C2C12 myoblasts. In contrast to the results from C2C12 cells, SMRT does not appear to play a role in adult skeletal muscle of whole body SMRT KO mice. Thus, SMRT plays a significant role in the differentiation of myoblasts.

## Introduction

With an aging population worldwide, the loss of muscle mass as found in sarcopenia is recognized as a significant medical problem [1–3]. Various qualitative changes in skeletal muscle, for example fibrosis and ectopic fatty deposition are being recognized as the cause of muscle loss in such disorders [4, 5]. Thus, significant work is focused on understanding muscle cell differentiation and muscle regeneration. These studies have demonstrated that a variety of transcription factors are involved in the process of both muscle differentiation and regeneration [6–11].

While nuclear receptor corepressors (CoRs) were thought to be a constitutional co-factor for nuclear receptors (NRs) in the regulation of target gene transcription, recent studies have shown that CoRs themselves play a larger role in the specific regulation of DNA transcription and act as gatekeepers of homeostasis [12–15]. Classically, in the absence of ligand, CoRs bind to target NRs and recruit a multi-protein complex that includes histone deacetylase 3 (HDAC3). In the formation of this complex, HDAC3 is activated and represses gene transcription. Recent studies have also shown that CoRs function in both presence or absence of ligand and that the amount of CoR present determines the sensitivity for the ligand in the target tissue [16–18].

Interestingly, each CoR has cell and nuclear receptor specific function. For example, NCoR1, a paralog of SMRT strongly suppresses the expression of TH receptor (TR)-target genes in the mouse liver while both SMRT and NCoR1 are required for the suppression of hepatic lipogenesis, indicating that each CoR can specifically function to regulate lipogenic genes likely through their recruitment by multiple NRs [16, 19, 20]. In skeletal muscle where NCoR1 has been deleted, mice demonstrate significant up-regulation of mitochondrial β-oxidation [21]. Interestingly, muscle specific deletion of HDAC3 leads to enhanced fatty acid catabolism rather than carbohydrate metabolism in the myocyte [22]. However, the muscle-specific function of SMRT, especially the role in the process of skeletal muscle development has not been elucidated.

By using a genome editing strategy, we made C2C12 SMRT knock-out cells (C2C12-SKO) to test the role of SMRT in a model of murine myocyte development. Herein we demonstrate

that these cells have significant up-regulation of fatty acid β-oxidation which coincides with the deterioration of myotube formation during the process of differentiation. Based on causal network analysis by proteomics, we also show that these cells have significant fibrosis consistent with up-regulation of TGF-β receptor-signaling. Taken together, we demonstrate that SMRT plays an important role as a strong suppressor of fatty acid catabolism and fibrosis during the process of skeletal muscle differentiation.

## Materials & methods

### Cell culture and differentiation of C2C12 cells

C2C12-WT and C2C12-SKO cells were cultured as previously described [23]. Briefly, these cells between passage 2–12 in growth medium were used for experiments. They were seeded at a density of 8000 cells/cm$^2$ in the 6-well plate, and cultured overnight in the antibiotic-free DMEM medium (high glucose) containing 10% fetal bovine serum (FBS) at 37°C in a humidified incubator containing 5% $CO_2$. In all experiments, cells were disseminated on the plates whose inside were coated by 1% gelatin solution (SIGMA-Aldrich) to adhere the cells on the bottom of each well. For differentiation into myotube formation, the culture medium was replaced with a differentiation medium (DMEM supplemented with 2% horse serum) for 5 days, and the medium was changed every couple of days.

### Mouse experiments

To generate the mouse strain with a post-natal global deletion of SMRT (SMRT$^{loxP/loxP}$UB-C$^{ERT2-Cre}$; UBC-SKO mice), we used the same strategy as previously described [20, 24, 25]. All experiments were approved by the Dokkyo Medical University Institutional Animal Care and Use Committee. 4–5 mice were housed in a cage in the animal facility with a 12 hours light/dark cycle and supplied with standard chow diet and water *ad libitum*. The experiments were performed in age- and sex-matched mice at 9–10 weeks of age unless otherwise specified. At that age, the UBC-SKO and SMRT$^{loxP/loxP}$ (control) animals were treated with tamoxifen (20mg/100g BW) for 5 days. At the end of the studies (at 24 weeks of age for both the UBC-SKO and control mice), all mice were euthanized via carbon dioxide inhalation. Blood and tissue samples were rapidly collected, and the level of recombination and extent of SMRT deletion was assessed by qPCR as reported previously [20, 25]. Only UBC-SKO mice with expression levels of the SMRT gene less than 90% of that were found in control animals were included in the experimental analysis.

### Genome editing strategy

The C2C12 SMRT-null clones (C2C12-SKO and C2C12-SKO2) were established by genome editing using a vector-based CRISPR-Cas9 system as described previously [23]. Briefly, to reduce off-target effects, we utilized the Cas9 double-nickase (Cas9n) method [26, 27]. The preparation of sgRNA-Cas9n expressing vectors was reported previously [23]. The sequences of sgRNA pairs were designed as shown in S1 Table. After co-transfection with an empty vector or a pair of SMRT exon2-targetting vectors were transfected using Lipofectamine 2000 (Thermo Fisher Scientific) and the cells were cultured in the presence of 1.5 μg/ml puromycin for 5 days. Surviving cells were then diluted and seeded as single colonies in each well of 96-well plate. Each clone was expanded and then screened for SMRT protein expression by western blotting using specific anti-SMRTe antibody (S1 Table). Genomic mutations in harvested clones were confirmed by DNA sequencing [23].

## si-RNA assays for Tgf-β isoforms and administration of a Tgf-β receptor inhibitor

C2C12-SKO cells were seeded in a 6-well plate and cultured in antibiotic-free DMEM medium containing 10% FBS overnight as described above. Next, the medium was changed to control medium and Stealth si-RNAs (Thermo Fisher Scientific) with a final concentration of 50 nM and prepared according to the manufacturer's protocol. Then, each of four si-RNA platforms; 50 nM si-Control, 50 nM si-Tgf-β1, 50 nM si-Tgf-β3, or 50 nM dual si-Tgf-βs (50 nM si-Tgf-β1 + 50 nM si-Tgf-β3) was added to the medium of the cells (S1 Table). For transfection, we used 9 μl of Lipofectamine RNAiMAX (Thermo Fisher Scientific) per well (on day 1 and day 3). At the end of day 4, treated cells were used for immunohistochemistry or harvested for expression analyses.

For the assay that used the Tgf-β receptor inhibitor; LY-364947 (Fujifilm Wako, 123–05981), C2C12-SKO cells were seeded and cultured overnight in antibiotic-free DMEM with 10% FBS in a 6-well plate (day 0). On day 1, the medium was changed to differentiation medium containing 10 μM of LY-364947 or vehicle. The cells were treated twice with LY-364947 or vehicle every 48 hours in the differentiation medium (day 1 and day 3). At the end of day 4, treated cells were used for immunohistochemistry or harvested for expression analyses.

## Immunofluorescence

Immunohistochemistry (IHC) studies were performed as previously described [23]. Briefly, cells were washed before and after fixation with 4% paraformaldehyde (PFA) (Fisher Scientific, Pittsburgh, PA, USA). The cells were incubated with the 1:300 dilution of anti-myosin heavy chain 4 (anti-MYH4) primary antibody in 5% skim milk at 4˚C, overnight (S1 Table). Secondary antibody; Alexa Fluor 488 goat anti-mouse IgG (H+L) and 4',6-diamidino-2-phenylindole (DAPI; Fujifilm Wako, Chemicals, Osaka, Japan 340–07971) (100 ng/ml) were used as manufacturer's instructions (S1 Table). Images were taken with a fluorescent microscope (Zeiss LSM 710, Oberkochen, Germany) using the program Slidebook 5.0 (Intelligent Imaging Innovations, Göttingen, Germany). The images were analyzed using ImageJ, and the fusion indexes were calculated as the percentage of the number of nuclei in MYH4-positive C2C12 myocytes divided by the total number of nuclei counted. Five fields were chosen randomly to measure the index [23, 28].

## Western blotting (WB)

Protein lysates from culture cells and mouse skeletal muscle for Western blot analysis were prepared as described previously [20, 23, 25]. Briefly, blots of 50 μg protein lysate for culture cell and 100 μg for mouse skeletal muscle were probed with the specific antibodies listed in S1 Table, according to the manufacturer's instructions (S1 Table). Appropriate horse-radish peroxidase (HRP)-conjugated antibodies were used as secondary antibodies (S1 Table). Then, ECL prime (GE Healthcare) was used for visualization. Protein density was quantified using Image Lab software (Bio-Rad).

For calculating relative expression, each protein level except SMRT, MYH4, and H4K5ac was normalized by the GAPDH expression level in the same lane respectively. Relative expressions of SMRT and MYH4 were normalized to the expression level of POL-2 protein. Relative expression of H4K5ac was normalized to the expression level of H4 protein. We assessed the relative expression of COL1A1 in Fig 3C normalized to GAPDH as we also probed for the FGF2 isoform which has a lower molecular weight. In contrast, we assessed COL1A1 in Fig 4C, 4F and S4C by POL-2 as higher molecular weight proteins were probed in the same blot.

## Fatty Acid Oxidation (FAO) rate

FAO rates in both C2C12-WT and C2C12-SKO cells were assessed by the fluorescence β-oxidation detection method using FAOBlue (Catalog number; FDV-0033, Funakoshi, Tokyo, Japan). Briefly, undifferentiated myoblasts of both cells were seeded at a density of $6x10^4$ cells/well in 24-well plates whose bottoms were coated by Cellmatirix Type I-C (Lot No. 191128, Nitta Gelatin, Osaka, JAPAN), and incubated for 24 hours as described above. Next, the cell culture medium was switched to serum-free DMEM which contained 5μM FAOBlue reagent and 1mM L-carnitine (Fujifilm Wako, Chemicals, Osaka, Japan 359–44361), followed by incubation at 37˚C for 3 hours. After cells were lysed in distilled water, the fluorescence of coumarin produced by β-oxidation of FAOBlue was measured by Varioskan® FLASH 4.00.53 (Thermo Fisher Scientific Inc., Waltham, MA, USA) with excitation at 405 nm and emission at 460 nm.

## Alkaline phosphatase (ALP) activity

ALP activity was assessed as an indication of osteoblastic differentiation in C2C12 myoblasts. The ALP activities of protein extracts from both C2C12-WT and C2C12-SKO myoblast cells were measured by p-nitrophenyl-phosphate method using LabAssay ALP™ALP (633–51021, FUJIFILM Wako, Gunma, JAPAN).

## Real-time quantitative PCR (RT-qPCR)

RNA extraction, cDNA synthesis and qPCR were performed as described previously [20, 25]. In the qPCR analyses, all mRNAs were quantified using Power SYBR Green PCR master mix (Roche) and the previously published primers listed in S1 Table in a total volume 10 μl (S1 Table). The relative mRNA levels of all genes were calculated using the standard-curve method and normalized to the level of GAPDH mRNA.

## Proteomics and pathway analysis

For sample preparation, three aliquots of cell lysate per group containing 10 μg of protein were prepared, and the steps of protein extraction and digestion were performed following the protocol of trypsin-based phase-transfer surfactants (PTS) buffer as described in Masuda T, et al. and Iwasaki M, et al. [29, 30]. Then, each sample was diluted by 50 μl of buffer A (0.1% formic acid, 5% acetonitrile), and subsequently desalted by GL-TipTM SDB (GL Sciences Inc., Tokyo, Japan) as prepared the manufacturer's protocol. The eluents were pooled and dried with vacuum centrifugation for following Liquid chromatography-tandem mass spectrometry (LC-MS/MS) analysis.

50 μl of protein sample diluted by buffer B (0.1% formic acid in acetonitrile) was loaded on an analytical column (0.3 ml Sc-Vial, 1030–51024, PTFE/Si-Sc-Cap 1030–51219, GL Sciences), and LC-MS/MS analysis was performed with the Triple TOF 6600 (AB Sciex, Framingham, MA), according to the protocol by Nakamura H, et al. [31].

For data analyses, the Sequential Window Acquisition of All Theoretical Mass Spectra (SWATH) were subsequently performed [31, 32]. The data of expression levels of proteins analyzed by SWATH was adjusted by peak intensity of β-Galactosidase (4333606, LC/MS Peptide Calibration Kit, PN 4465867, Sciex). The proteins whose expression levels acquired from C2C12-SKO cells changed more than 2- or 0.5-fold compared to C2C12-WT cells were used as specific proteins (S2 Table).

The data list of differentially expressed proteins was subsequently imported to Ingenuity Pathway Analysis (IPA) software (Qiagen, Hilden, Germany) for constructing both pathways and networks between protein molecules. The statistical method calculating for activation z-

score in the analyses of both canonical pathway and the upstream and downstream (diseases and functions) analyses were based on the methods previously reported [33, 34].

### Statistical analyses

Statistical analysis was performed using the Prism, Ver.7 program (GraphPad software, San Diego, CA, USA). All data (gene expression and quantifications of protein amount) were presented as mean ± standard error of the mean (SEM). These data were normalized by the values of a control cell (C2C12-WT, C2C12-SKO with vehicle) or of a control mouse group (SMRT$^{loxP/loxP}$). Significance in the difference between two groups (mRNA levels, quantifications of protein amount, fusion index, FAO rate, ALP activity, and mouse quadriceps weight) were determined by the unpaired Student's t-test. The significance in the difference between three groups (mRNA levels, quantifications of protein amount, and fusion index) were determined by Repeated Measures One-way analysis of variance (ANOVA) with the Tukey-Kramer *post hoc* test. The significances of cellular mRNA and protein levels in C2C12-WT compared between the series of incubation time point were also analyzed by Repeated Measures One-way ANOVA with the Tukey-Kramer *post hoc* test.

## Results

### SMRT regulates myotube formation in the differentiation process of C2C12 cells

The function of SMRT during the differentiation process of skeletal muscle has not been elucidated. To examine SMRT expression in muscle development, we used mouse C2C12 wild type (C2C12-WT) cells as a model. At first, we performed expression analyses for both SMRT mRNA and protein in a series of time points in the differentiation process, and confirmed that myotubes were formed in C2C12-WT cells that was cultured in the control medium for 5 days. As shown in S1A Fig, the transcription of *Smrt* mRNA was significantly up-regulated in the former on days 1–2 and the level remained stable through days 3–5 (S1A Fig). On the other hand, the protein expression level of SMRT increased by day 5, indicating that SMRT protein accumulates in the latter differentiation phase of C2C12-WT cells (S1B Fig). These data indicate that the function of SMRT seems to be more important in the later stage of muscle differentiation.

To elucidate SMRT's role in skeletal muscle differentiation, we made two clones of SMRT-null C2C12 cells which were named C2C12-SMRT KO and C2C12-SMRT KO2 (C2C12-SKO and C2C12-SKO2) utilizing a genome-editing strategy. Immunohistochemistry (IHC) analysis using an anti-MYH4 antibody demonstrates that both C2C12-SKO and C2C12-SKO2 cells show immature myotubes compared with C2C12-WT cell (Figs 1A and S1C). Consistent with the result of IHC, expression of *Myosin heavy chain (Myh4)* was lower in both mRNA and protein levels in the two SMRT-null clones (Figs 1B, 1C, S1D and S1E). However, further gene analyses showed that the majority of myogenic markers tested in the C2C12-SKO cells showed higher expression patterns, indicating that SMRT seems to effect specific myogenic target genes such as *Myh4* in the differentiation process (Figs 1B, 1C, S1D and S1E). Taken together, two clones of SMRT-null cell showed a similar phenotype which impairs normal myotube formation. These data suggest that SMRT seems to play an important role in the regulation of specific myogenic gene expression in the differentiation process of C2C12 cells.

### SMRT suppresses β-oxidation during C2C12 cell differentiation

Next, we performed further analyses using one of the SMRT-null clones (C2C12-SKO) and investigated the mechanism by which SMRT regulates metabolic genes during muscle

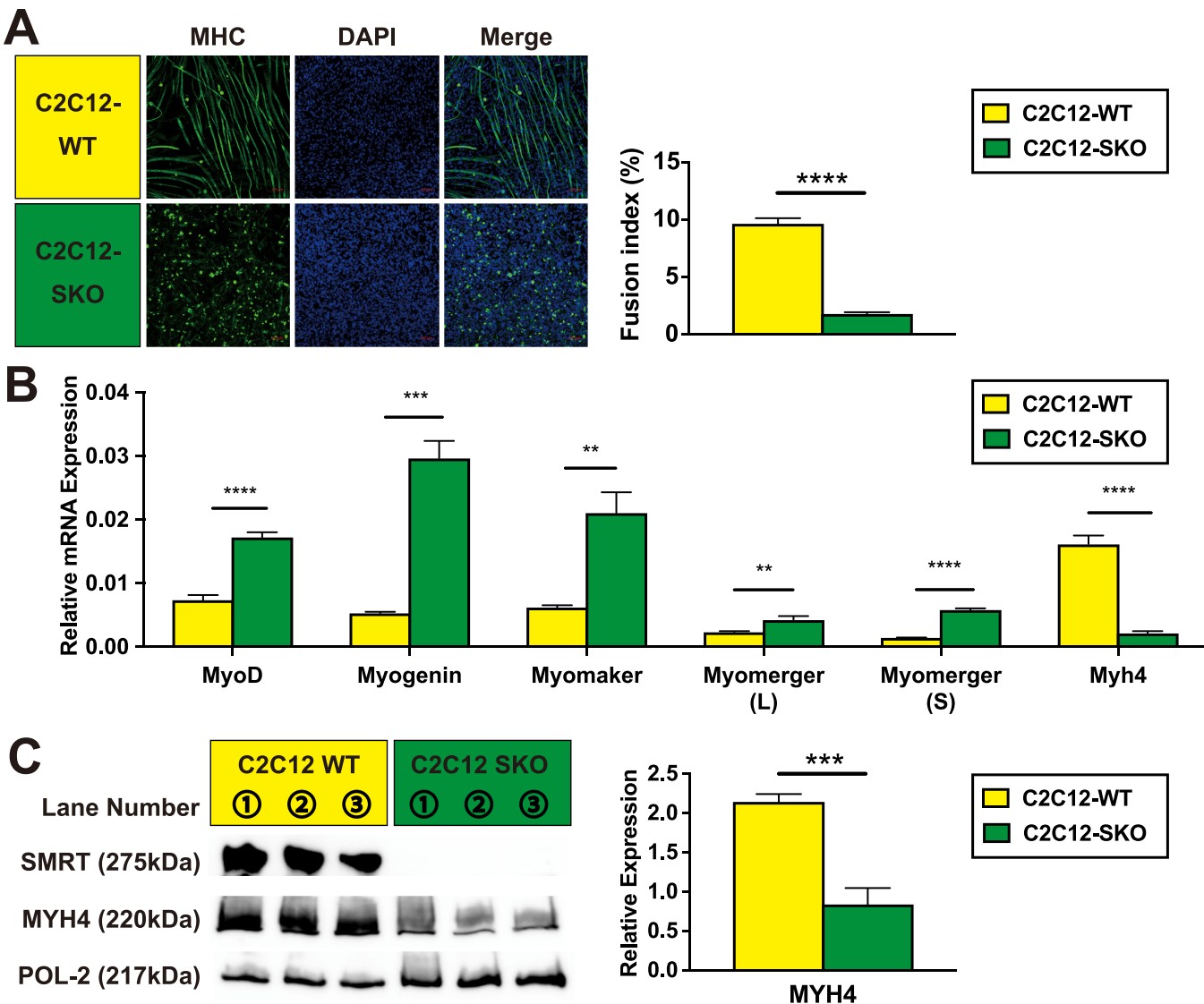

**Fig 1. SMRT regulates myotube formation in the differentiation process of C2C12 cells.** (A) Both C2C12-WT and C2C12-SKO cells were induced to differentiate in differentiation medium. The expression of myosin heavy chain 4 (MYH4) in both C2C12-WT and C2C12-SKO cells were assessed by IHC. Myotubes were stained by anti-MYH4 antibody (Green). Nuclei were stained by DAPI (Blue). The fusion indexes of the cells were calculated from IHC. (B) mRNA expression of myogenic-related genes in both C2C12-WT and C2C12-SKO cells were quantified by qPCR (n = 3 per group). (C) Protein expression of SMRT and MYH4 in both C2C12-WT and C2C12-SKO cells were assessed by WB (n = 3 per group). Student t-tests were used for statistical analyses in the calculation for fusion index and all qPCR and quantification of protein expression. Results are shown as the mean±SEM (error bars represent SEM), and the p-value are shown as ****p<0.0001, ***p<0.001, **p<0.01.

differentiation. Previous studies which examined the corepressor complex in muscle (muscle specific NCoR1KO and the global HDAC3 mutant) showed that both models have significant impact on oxidative metabolism in skeletal muscle [21, 22]. At first, we investigated mRNA and protein levels of β-oxidation-related genes in the pre-differentiation stage of both C2C12-WT and C2C12-SKO cells. While the mRNA level of *Pgc-1α*; a potent transcriptional factor of muscle β-oxidation was significantly lower the nuclear receptor, *Pparδ*, was significantly up-regulated in C2C12-SKO cells (S2A Fig). Contrary to these results, protein expression of both PPARδ and PGC-1α were significantly up-regulated in C2C12-SKO cells compared with C2C12-WT cells (S2A and S2B Fig). As β-oxidation-related-genes were up-

regulated in C2C12-SKO cells even before differentiation, the alterations of these genes seems to reflect the difference in β-oxidation between C2C12-WT and C2C12-SKO cells. Indeed, as shown in the S2C Fig, we confirmed that the fatty acid oxidation (FAO) rate of C2C12-SKO cells was significantly higher than that of WT cells, indicating that β-oxidation is up-regulated in SMRT-null myoblasts even before myotube differentiation (S2C Fig). Based on this, we assumed that SMRT may control energy metabolism during muscle differentiation. Both mRNA and protein expressions of *Pparδ and Pgc-1α* in the post-differentiation state showed similar patterns to those of pre-differentiation state in C2C12-SKO cells (Fig 2A and 2B). While the mRNA expression of *Pgc-1α* was significantly lower the protein level of PGC-1α expression was higher in the C2C12-SKO cells, indicating that high amounts of PPARδ protein

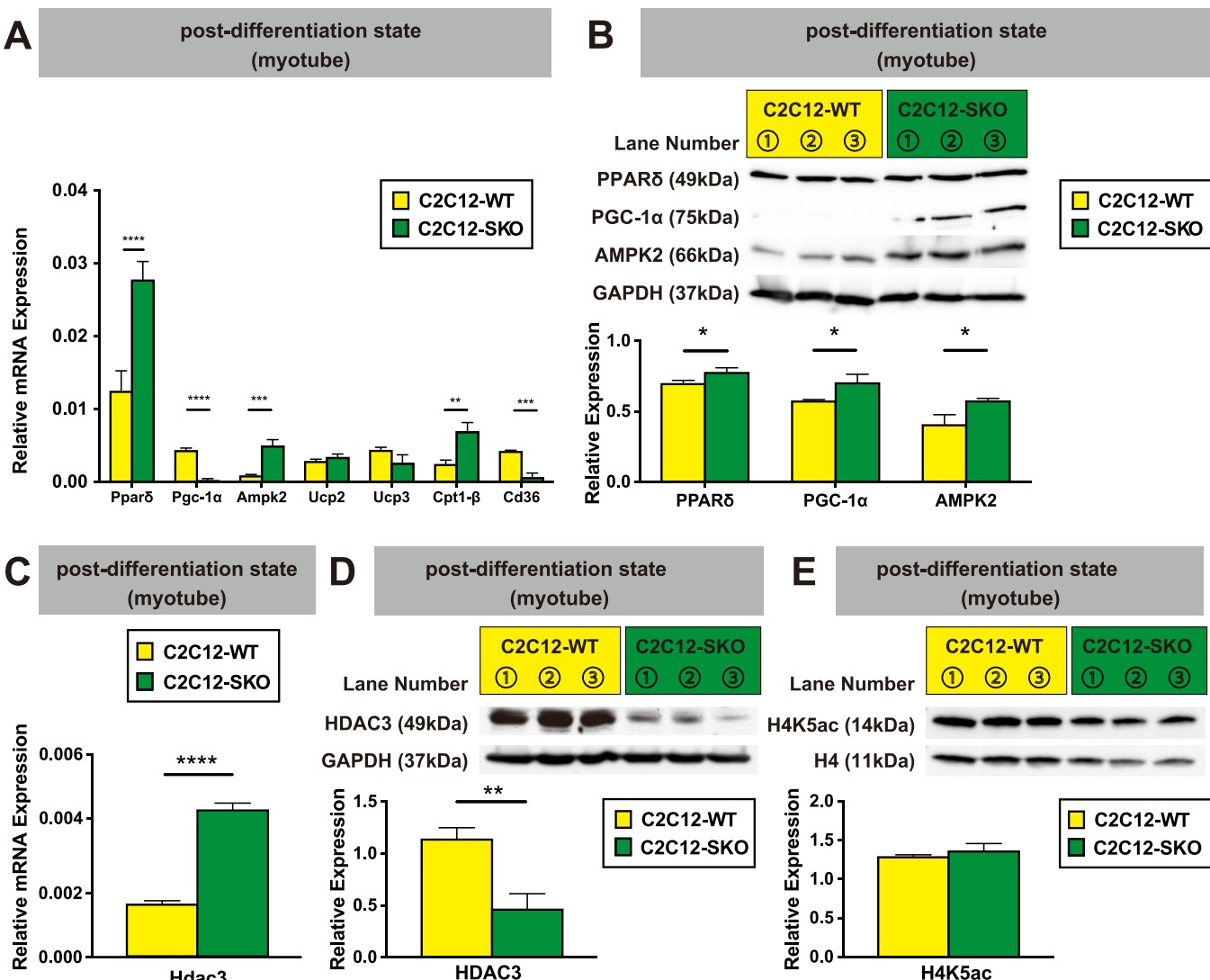

**Fig 2. SMRT suppresses β-oxidation during the differentiation of C2C12 cells.** (A) The mRNA expressions of β-oxidation-related gene in both C2C12-WT and C2C12-SKO cells were quantified by qPCR (n = 3 per group). (B) Protein expressions of PPARδ, PGC-1α and AMPK2 were assessed by WB (n = 3 per group). (C) The mRNA expression of *Hdac3* in both C2C12-WT and C2C12-SKO cells was quantified by qPCR (n = 3 per group). (D) Protein expression of HDAC3 was assessed by WB (n = 3 per group). (E) Protein expressions of histone H4K5ac and H4 were assessed by WB (n = 3 per group). Student t-tests were used for statistical analyses in all qPCR and quantification of protein expression. Results are shown as the mean±SEM (error bars represent SEM), and the p-value are shown as ****p<0.0001, ***p<0.001, **p<0.01, and *p<0.05.

may protect PGC-1α from degradation induced by poly-ubiquitination (Fig 2A and 2B) [35]. Although the mRNA expressions of *Uncoupling protein2 and 3 (Ucp2, Ucp3)* in C2C12-SKO cells were comparable to control, as we expected, both mRNA and protein levels of *AMP-activated protein kinase2 (Ampk2)* and mRNA level of *Carnitine palmytoyltransferase 1-β (Cpt1-β)* were significantly up-regulated in C2C12-SKO cells (Fig 2A and 2B). While mRNA expression of *Cd36* was decreased during differentiation as seen in Fig 2A, these results taken together the data suggest that the intercellular β-oxidation level was significantly elevated through the process of differentiation of C2C12-SKO cells (Fig 2A and 2B).

HDAC3 is known as the major conserved partner of NCoR1/SMRT in the formation of the CoR complex that elicits enzymatic activity as a histone deacetylase to induce transcriptional repression [18, 19, 36]. Previous studies using a muscle specific HDAC3-null mouse or NCoR1/SMRT double mutant mice which lack a HDAC3-binding domain manifest a similar muscular phenotype. For example, as HDAC3 activity fell the catabolism of lipids became more predominant than that of carbohydrates in the skeletal muscle of these mouse models [22, 37, 38]. To confirm the net effect of SMRT deletion on HDAC3, we investigated its expression. While the mRNA level of *Hdac3* was up-regulated the protein level was lower in C2C12-SKO cells, indicating that C2C12-SKO cells may have lower HDAC3 activity compared with wild type cells (Fig 2C and 2D).

To investigate the effect of SMRT deletion on its paralog NCoR1, we performed expression analyses of both its mRNA and protein levels in the post-differentiation state in C2C12-SKO cells. As shown in S2D and S2E Fig, both were up-regulated in C2C12-SKO cells (S2D and S2E Fig). However, despite the higher expression of NCoR1 in C2C12-SKO cells, the protein levels of acetylated histone H4K5 (H4K5ac) in C2C12-SKO were comparable to those of C2C12-WT cells (Fig 2E). Taken these data together, SMRT seems to be major suppressor of β-oxidation in the differentiation process of C2C12 cells.

## SMRT suppresses Tgf-β receptor signaling-mediated fibrosis in C2C12 cells

To further clarify the mechanism of how SMRT regulates myotube formation in C2C12 cells, we performed a comprehensive proteomics analysis using LC-MS/MS comparing fully differentiated C2C12-WT and C2C12-SKO cells. We identified 397 differentially expressed proteins in C2C12-SKO cells, and the statistical significance was found in the expression level of 85 of these proteins, with 44 proteins significantly up-regulated, and 41 proteins significantly down-regulated (S2 Table, S3A Fig). Interestingly, some connective tissue-related proteins, for instance, Phosphatidylglycerophosphate Synthase 1 (PGS1), Collagen Type I Alpha I (COL1A1), Sarco/Endoplasmic Reticulum Ca$^{2+}$-ATPase (SERCA), and Troponin I were up-regulated while glycolysis-related enzymes such as Hexokinase1 (HXK1), Enorase3 (ENOB), and Phosphoglycerate Kinase1 (PGK1) were significantly down-regulated. This suggests that the deletion of SMRT causes both constitutive and catabolic changes in C2C12-SKO cells. To elucidate the mediator of these changes in C2C12-SKO cells, we analyzed their expression profile using Ingenuity Pathways Analysis [33]. Top canonical analysis showed that glucose catabolism was enhanced in C2C12-SKO cells (e.g. Glycolysis, Gluconeogenesis), and TGFB1 and SMAD were identified as top-ranked up-stream regulators, indicating that Tgf-β signaling seems to be significantly up-regulated in C2C12-SKO cells (S3B and S3C Fig).

To confirm that SMRT deletion did affect transcriptional derepression of Tgf-β signaling, we performed mRNA expression analyses of the Tgf-β family. Indeed, the majority of Tgf-β isoforms (*Tgf-β1*, *Tgf-β2*, and *Tgf-β3*) and another Tgf-β family member, *Bone morphogenetic protein4 (Bmp4)* were significantly up-regulated in both the pre-differentiation and post-differentiation state in C2C12-SKO cells (S3D Fig, Fig 3A). Further analyses demonstrated that

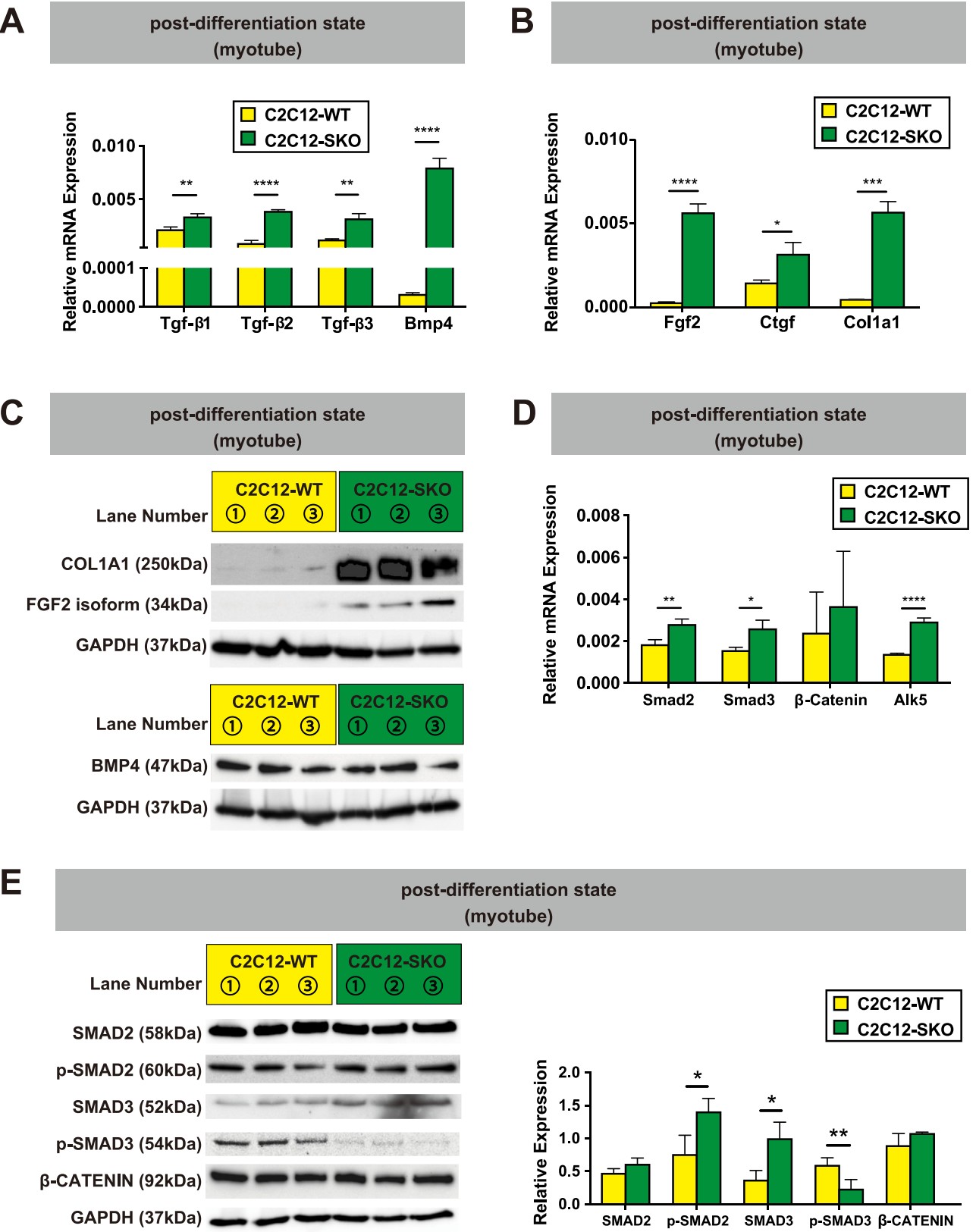

**Fig 3. SMRT suppresses Tgf-β receptor signaling-mediated fibrosis in C2C12 cell.** (A) The expression of Tgf-β signaling-related genes in both C2C12-WT and C2C12-SKO cells were quantified by qPCR (n = 3 per group). (B) The expression of fibrosis-related genes in both C2C12-WT and C2C12-SKO cells were quantified by qPCR (n = 3 per group). (C) Protein expression of markers of fibrosis and BMP4 in both C2C12-WT and C2C12-SKO cells were assessed by WB (n = 3 per group). (D) The expression of Tgf-β receptor-target genes in both C2C12-WT and C2C12-SKO cells were quantified by qPCR (n = 3 per group). (E) The protein levels of transcriptional mediators in Tgf-β receptor signaling were assessed by WB in both C2C12-WT and C2C12-SKO cells. (n = 3 per group). Student t-tests were used for statistical analyses in all qPCR and quantification of protein expression. All results of bar graph in this figure are shown as the mean±SEM (error bars represent SEM), and the p-value are shown as ****$p<0.0001$, ***$p<0.001$, **$p<0.01$, and *$p<0.05$.

the mRNA expression of the fibrosis-related genes, *Col1a1*, *Fgf2*, and *Connective tissue growth factor (Ctgf)* which are recognized as target genes of Tgf-β signaling showed higher expression in the post-differentiation state in C2C12-SKO cells (Fig 3B). Consistent with this, the protein levels of COL1A1 and FGF2 were also increased in C2C12-SKO cells, indicating that SMRT deletion seemed to increase markers of fibrosis in C2C12-SKO cells (Fig 3C). While the mRNA level of BMP4 was higher, the protein level in C2C12-SKO cells was comparable to C2C12-WT cells (Fig 3C). To confirm the possibility of osteoblastic differentiation caused by enhanced BMP4 signaling in the pre-differentiation state of C2C12 myoblast cells, we performed further analysis for ALP activities compared between C2C12-WT and C2C12-SKO cells as osteoblast formation accompanied with high ALP activity has been shown to be strongly induced by BMP4 treatment in C2C12 myoblasts [39]. As shown in S3E Fig, indeed, C2C12-SKO cells had significantly higher ALP activity compared with wild type in the pre-differentiated state (S3E Fig). This indicates that, while the protein level of BMP4 in C2C12-SKO cells was comparable to that of WT cells, the osteoblastic change caused by the activation of BMP4 signaling may also play a role in the deterioration of myotube formation seen in C2C12-SKO cells (Figs 3C and S3E).

To clarify the mechanism of enhanced signaling resulting in pathways activated in fibrosis, we focused on transcriptional levels of Tgf-β receptor signaling mediators [40–42]. As shown in Fig 3D, mRNA expression of *Smad2* and *Smad3*, which are representative transcriptional activators of Tgf-β receptor signaling in C2C12-SKO cells were increased and *Activin A receptor type-II-like kinase 5 (Tgf-β receptor type I*, *Alk5)* was also significantly up-regulated in C2C12-SKO cells. In contrast, both mRNA and protein expression of *β-Catenin* in C2C12-SKO cells were comparable to wild type cells, indicating that the activation of Tgf-β receptor signaling does not change the amount of β-CATENIN which is the major cofactor of PPARδ-mediated NR complex (Fig 3D and 3E) [43, 44]. Furthermore, while the expression level of the phosphorylated form of SMAD3 (p-SMAD3), a pivotal transcriptional mediator of Tgf-β receptor signaling was significantly lower, the expression level of non-phosphorylated SMAD2 was not decreased and p-SMAD2 was up-regulated in C2C12-SKO cells. These data suggest that fibrotic changes may be activated by multiple mediators downstream of Tgf-β receptor signaling via both transcriptional and translational regulation (Fig 3D and 3E) [28, 45].

## SMRT plays an important role in both myotube formation and protection against fibrosis during differentiation of C2C12 cells via Tgf-β receptor signaling

To clarify SMRTs function in the context of Tgf-β receptor signaling, we investigated the effect of small interfering RNA (si-RNA) on Tgf-β receptor isoforms during C2C12 cell differentiation. As shown in Fig 4A, IHC demonstrates that C2C12-SKO cells treated with si-Tgf-β1 showed higher expression of MYH4 protein and the improvement of the fusion index compared to si-Control, indicating the recovery of myotube formation (Fig 4A). As expected, the mRNA level of *Tgf-β1* was lower in the C2C12-SKO treated by si-Tgf-β1, the protein level of

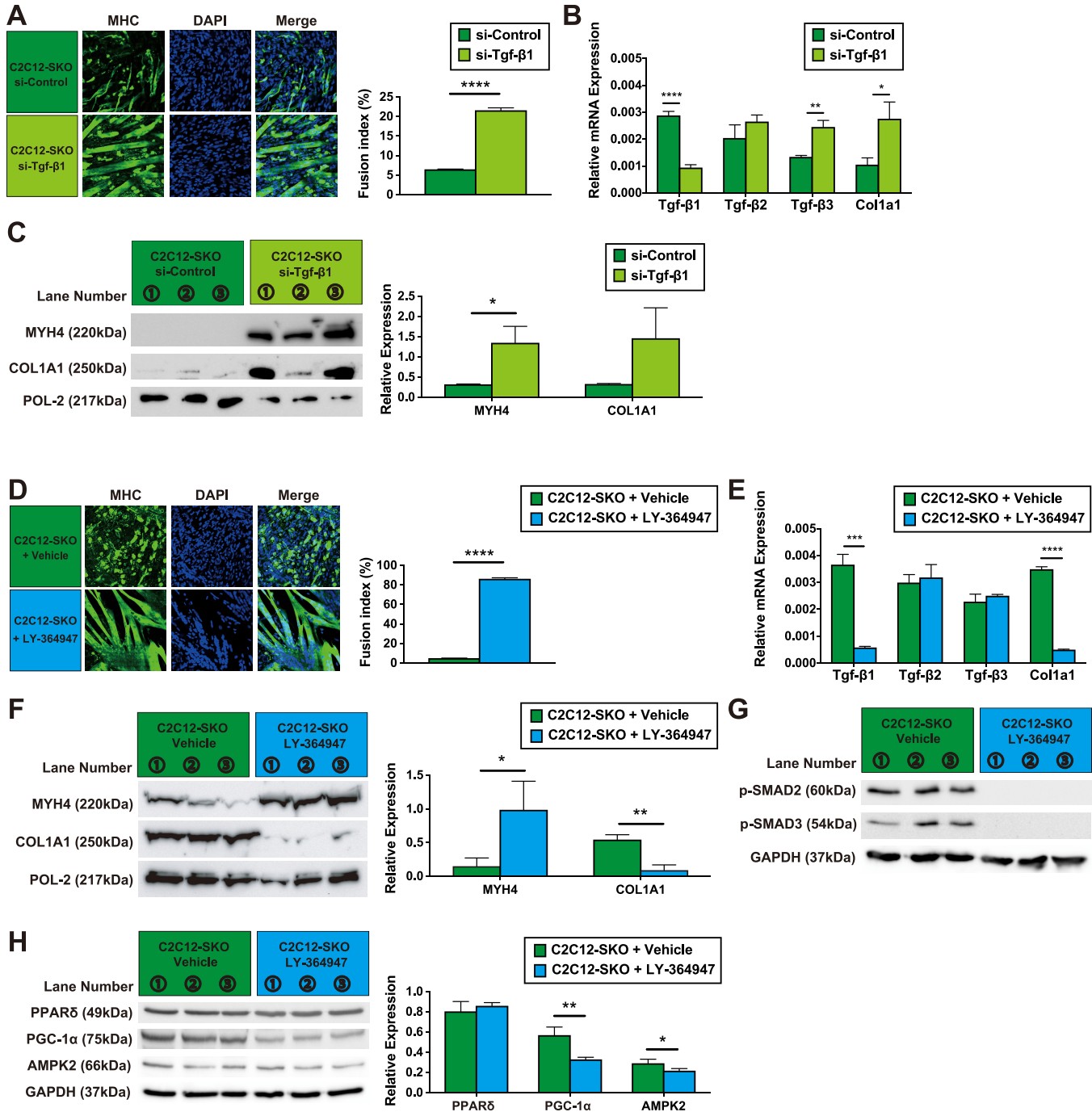

**Fig 4. SMRT plays an important role in both myotube formation and protection against fibrosis during the differentiation of C2C12 cells via Tgf-β receptor signaling.** (A) Two groups of C2C12-SKO cells treated with either si-Tgf-β1 or si-Control were induced to differentiate in control medium for 4 days. The expressions of MYH4 in both cells were assessed by IHC. Myotubes were stained by MYH4 (Green). Nuclei were stained by DAPI (Blue). (B) mRNA expressions of Tgf-β isoforms and *Col1a1* in two groups of C2C12-SKO cells treated with either si-Tgf-β1 or si-Control were quantified by qPCR (n = 3 per group). (C) Protein expressions of MYH4 and COL1A1 in two groups of C2C12-SKO cells treated with either si-Tgf-β1 or si-Control were assessed by WB (n = 3 per group). (D) Two groups of C2C12-SKO cells treated with either vehicle or LY-364947 were induced to differentiation in control medium for 4 days. The expressions of MYH4 in both cells were assessed by IHC. Myotubes were stained by MYH4 (Green). Nuclei were stained by DAPI (Blue). (E) mRNA expressions of Tgf-β isoforms and *Col1a1* in two groups of C2C12-SKO cells treated with either vehicle or LY-364947 were quantified by qPCR (n = 3 per group). (F) Protein expressions of MYH4 and COL1A1 in two groups of C2C12-SKO cells treated with either vehicle or LY-364947 were assessed by WB (n = 3). per group). (G) Protein expressions of p-SMAD2 and p-SMAD3 in two groups of C2C12-SKO cells treated with either vehicle or LY-364947 were assessed by WB (n = 3 per group). (H) Protein expressions of PPARδ, PGC-1α, and AMPK2 in two groups of C2C12-SKO cells treated with either vehicle or LY-364947 were assessed by WB (n = 3 per group). Student t-tests were used for statistical analyses in all qPCR and quantification of protein expression. Results are shown as the mean±SEM (error bars represent SEM), and the p-value are shown as $^{****}p<0.0001$, $^{***}p<0.001$; $^{**}$, $p<0.01$; $^{*}$, $p<0.05$.

MYH4 was significantly up-regulated compared to the C2C12-SKO cells treated by si-Control (Fig 4B and 4C). However, treatment with si-Tgf-β1 did not lower the marker of fibrosis, *Col1a1*, in C2C12-SKO cells. (Fig 4B and 4C). Interestingly, *Tgf-β3* was significantly up-regulated in C2C12-SKO cells treated with si-Tgf-β1(Fig 4B). Consistent with this, the protein level of COL1A1 had a trend of higher expression in the C2C12-SKO cells treated with si-Tgf-β1, compared with the cells treated by si-Control (Fig 4C). These data suggest that the reciprocal up-regulation of other Tgf-β isoforms may promote fibrosis in C2C12-SKO cells.

To further analyze this, we performed further si-RNA experiments targeting another Tgf-β receptor isoform; *Tgf-β3*, and a dual siRNA targeting both *Tgf-β1* and *Tgf-β3* (si-Tgf-β(1+3)). As shown S4A Fig, myotube formation recovered in both the C2C12-SKO cells treated by either si-Tgf-β3 or the dual si-Tgf-β(1+3) (S4A Fig). Intriguingly, the C2C12-SKO cells treated with si-Tgf-β3 showed lower mRNA expression of *Tgf-β1*, and the dual si-Tgf-β(1+3) siRNA treatred cells showed significantly lower expressions of all three isoforms. *Tgf-β2* seemed to have little effect on the myoblast differentiation program as its expression did not show reciprocal up-regulation (S4B Fig). However, the mRNA expression of *Col1a1* in both C2C12-SKO cells treated with si-Tgf-β3 and dual si-Tgf-β(1+3) siRNAs were comparable to that of si-Control (S4B Fig). Consistent with this, both C2C12-SKO cells treated with si-Tgf-β3 and dual si-Tgf-β(1+3) siRNAs showed a trend to recovery of the MYH4 protein. However, the expression of COL1A1 in both remained higher compared with the cells treated with si-Control. This suggests the possibility that the effect of siRNA for each Tgf-β isoform was not strong enough to prevent fibrotic change in C2C12 cells (S4C Fig).

To examine this, we investigated the effect of LY-364947, a competitive inhibitor for the Tgf-β receptors on the differentiation process of C2C12 cells. As expected, C2C12-SKO cells treated with LY-364947 showed increased myotube formation by IHC, and significantly higher expression of MYH4 protein by WB (Fig 4D and 4F). The gene profiles analyzed by qPCR demonstrated that only *Tgf-β1* mRNA was significantly down-regulated in C2C12-SKO cells treated with LY-364947, and other Tgf-β isoforms (*Tgf-β2* and *Tgf-β3*) in the cell were comparable to the controls (C2C12-SKO cell treated with vehicle) (Fig 4E). Although the administration of LY-364947 significantly decreased the expression of only Tgf-β1, the fibrotic marker; *Col1a1* was significantly reduced in C2C12-SKO cells (Fig 4E). Consistent with this, protein levels of COL1A1 were also significantly down-regulated in C2C12-SKO cells treated with LY-364947 (Fig 4F).

To assess the net effect of Tgf-β signaling both on the myogenic differentiation and β-oxidation-mediated lipid metabolism, we evaluated the protein expression of both p-SMADs and β-oxidation-related genes between two group of C2C12-SKO cells treated by vehicle or LY-364947. Indeed, as shown in Fig 4G, the protein expression of both p-SMAD2 and p-SMAD3 were decreased in C2C12-SKO cells treated by LY-364947 compared with the cells treated by vehicle. This indicates that enhanced Tgf-β receptor signaling in C2C12-SKO cells has a suppressive effect on myotube formation (Fig 4G). Next, we also evaluated the protein expression of β-oxidation-related genes between the two cell groups. While the relative expression of PPARδ showed no difference between the two cell groups, interestingly, PGC-1α and AMPK2 were significantly down-regulated in C2C12-SKO cells treated with LY-36494, suggesting the possibility that suppression of Tgf-β signaling leads to a decrease in β-oxidation-mediated lipid metabolism (Fig 4H). Taken together, these data suggest that while the inhibition of each Tgf-β isoform would be important for normal myofiber formation the complete blockade of Tgf-β receptor signaling is pivotal for preventing fibrosis as well as suppressing the expression of specific β-oxidation-related genes during the differentiation of C2C12 cells.

## Post-natal SMRT deletion has minimal effects on both β-oxidation and Tgf-β signaling in *in vivo* mouse skeletal muscle

To confirm SMRTs function in mature mouse skeletal muscle *in vivo*, we used a SMRT deletion strategy using tamoxifen-inducible ubiquitin C (UBC) ERT2-Cre recombinase to avoid the embryonic lethality associated with global SMRT deletion [20, 25, 46]. We previously demonstrated that UBC-SKO mice had lower skeletal muscle mass which associates with significant obesity [20]. As shown in Fig 5A, SMRT protein was significantly decreased in the skeletal muscle of whole-body SMRT KO mice (UBC-SKO) compared with SMRT$^{loxP/loxP}$ mice without ERT2-Cre (Control) (Fig 5A). As expected, UBC-SKO mice have a smaller sized quadriceps with significantly lower weight than controls (S5A and S5B Fig).

To investigate the metabolic effects in skeletal muscle of UBC-SKO mice, we performed expression analyses of β-oxidation-related genes, compared with control. Gene profiling by qPCR demonstrated that mRNA levels of all β-oxidation-related genes (ex. *Pparδ*, *Pgc-1α*, and *Ampk2*) tested in UBC-SKO skeletal muscle were comparable to control (Fig 5B). While protein levels of PPARδ were significantly lower in UBC-SKO mice compared with control, the levels of PGC-1α were similar, indicating post-natal SMRT deletion caused few changes in β-oxidation in fully-developed skeletal muscle (Fig 5C).

We next further analyzed the effects of SMRT deletion on both Tgf-β signaling and fibrotic markers in skeletal muscle. As shown in Fig 5D, although *Tgf-β1* and *Bmp4* were comparable to those of controls, *β-Catenin* and other Tgf-β isoforms (ex. *Tgf-β2* and *Tgf-β3*) were

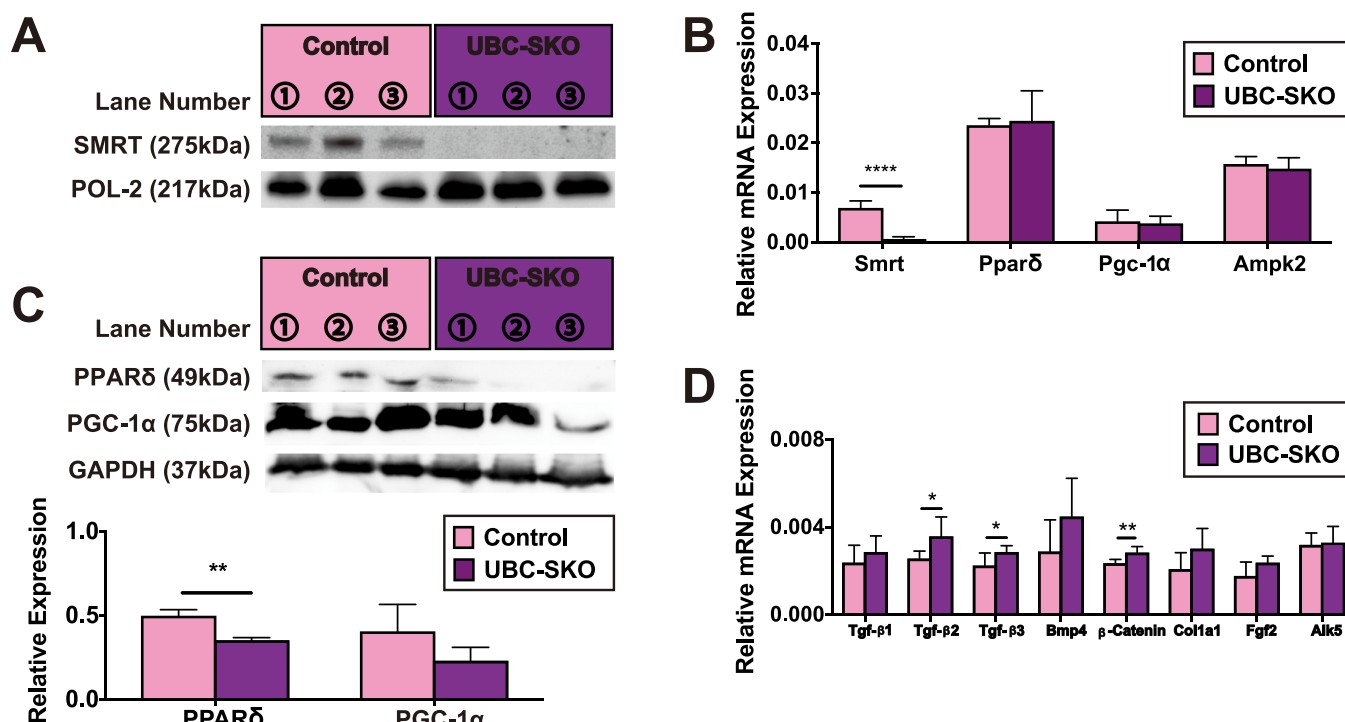

**Fig 5. Post-natal SMRT deletion has minimal effects on both β-oxidation and Tgf-β signaling in *in vivo* mouse skeletal muscle.** (A) Protein levels of SMRT in the skeletal muscle of both UBC-SKO and control mice was assessed by WB (n = 3 mice/group). (B) The expressions of *Smrt* and β-oxidation-related genes in the skeletal muscle of both UBC-SKO and control mice were quantified by qPCR (n = 5–6 mice/group). (C) Protein expressions of PPARδ and PGC-1α in the skeletal muscle of both UBC-SKO and control mice were assessed by WB (n = 3 mice/per group). (D) The expressions of Tgf-β receptor signaling-related genes and markers of fibrosis in the skeletal muscle of both UBC-SKO and control mice were quantified by qPCR (n = 5–6 mice/group). Student t-tests were used for statistical analyses in all qPCR and quantification of protein expression. All results are shown as the mean±SEM (error bars represent SEM), the p-value was shown as; ****, p< 0.0001; **, p< 0.01; *, p< 0.05.

significantly increased in the skeletal muscle of UBC-SKO mice (Fig 5D). However, the mRNA expressions of markers of fibrosis (ex. *Col1a1* and *Fgf2*) and Tgf-β receptor signaling-related genes (eg. *Alk5*) were not different between controls and UBC-SKO mouse muscle (Fig 5D). Taken together, these results imply that while there may still be the possibility that some certain Tgf-β receptors are activated after the post-natal deletion of SMRT this is not enough to change β-oxidation or fibrosis in the skeletal muscle of the UBC-SKO mice.

## Discussion

Degenerative muscle disorders originating from congenital or acquired disease directly causes serious restrictions on the activities of daily living (ADL) for patients. In addition, because society as a whole is aging, sarcopenia is increasingly recognized as a world-wide medical problem [3]. Based on this, research focusing on muscle development is an area of intense interest. Importantly it is critical to understand the gene regulatory mechanisms involved during the process of development or regeneration [7, 9, 10, 47].

Both SMRT and NCoR1 are known as the representative orthologs of CoRs which are broadly expressed in major organs in mice and humans. Previous studies have demonstrated that the each CoR plays a tissue-specific role in the regulation of DNA transcription [12, 48–50]. Indeed, using genetic strategies, recent studies have demonstrated that muscle-specific deficiency of either NCoR1 or HDAC3 in mice can cause the up-regulation of fatty acid β-oxidation [21, 22]. However, the role of SMRT in skeletal muscle has not been elucidated yet.

To investigate the function of SMRT in the mouse skeletal muscle, we made SMRT-null clones from mouse C2C12 myoblast cells using CRISPR-Cas9. Interestingly, the loss of SMRT in these cells led to the up-regulation of genes important in fatty-acid β-oxidation which is usually activated by physical exercise in skeletal muscle. We also found that these cells had a significant loss of myofiber formation (Figs 1A and S1C).

Previous studies demonstrated that the pivotal transcriptional mediator; PGC-1α strongly regulates fatty acid β-oxidation via the nuclear receptor PPARδ in skeletal muscle [35]. Thus we first focused on SMRTs function in the context of the regulation of fatty acid β-oxidation regulation in C2C12 cells also in part because PPARδ was previously reported as a binding partner for SMRT [51]. Indeed, we found that C2C12-SKO cells had high protein expression levels of PPARδ and PGC-1α in both the pre- and post-differentiation state (Figs 2B and S2B). To confirm the SMRT-null effect in adult skeletal muscle, we also analyzed the muscle tissue dissected from global SMRT KO (UBC-SKO) mice and did not see similar elevation of PPARδ and PGC-1α, which was inconsistent with the results observed in C2C12-SKO myotubes (Figs 2B and 5C). These contradictions may be caused by the difference in the function of SMRT before and after the muscle tissue development. Indeed, while the embryonic global SMRT deletion causes embryonic lethality because of congenital cardiac defects we previously confirmed that adult UBC-SKO mice have normal cardiac function [20, 46]. Taken together, these data suggest that SMRT works as a master regulator for lipid catabolism during myocyte differentiation in C2C12 cells. In the adult mouse after development has occurred, SMRT does not appear to play a role in this process.

Further comprehensive proteomics analysis using LC-MS/MS and following IPA pathway analysis demonstrated that C2C12-SKO cells had lower expression of glycolysis-related enzymes. This suggests that the cells seemed to have a catabolic shift from carbohydrates to fatty acids which was a similar phenotype seen in the genetic mouse models reported previously (S2 Table, S3B Fig) [22, 37, 38]. However, we also found that Tgf-β receptor signaling is the predominant pathway activated in C2C12-SKO cells. Both Tgf-β family members and fibrosis markers were significantly up-regulated in the C2C12-SKO cells (Figs 3A, 3B, 3D and

S3D). Tgf-βs have been known as the signaling mediators which cause fibrosis in cardiac or skeletal muscle, leading to degenerative muscle disorders such as heart failure and sarcopenia [28, 52]. Indeed, the inhibition of Tgf-β receptor signaling could induce myotube formation and ameliorate fibrosis in the C2C12-SKO cells (Fig 4D–4F). Of note, we found that while the direct inhibition of Tgf-β signaling by siRNA could re-establish myotube formation it did not improve markers of fibrosis in C2C12-SKO cells (Figs 4A–4C and S4A–S4C). Thus the regulation of Tgf-β receptor signaling by the SMRT/HDAC3 repressor complex may predominantly effect the inhibition of fibrosis rather than the process of myofiber differentiation in C2C12 cells which was elicited by each Tgf-β receptor isoform. While the protein level of p-SMAD3 was lower than the other regulators of Tgf-β receptor signaling non-phosphorylated SMAD3 and p-SMAD2 were significantly up-regulated in the C2C12-SKO cells (Fig 3E). Indeed, further evaluation of the C2C12-SKO cells treated by LY-364947 showed significant lower expression level of both p-SMADs (p-SMAD2, 3) and some certain β-oxidation-related genes (PGC-1α, and AMPK2). Thus, Tgf-β receptor signaling suppresses myotube formation and β-oxidation-mediated lipid metabolism in C2C12 cells (Fig 4G and 4H). Of note, Fig 4F and 4G indicate that the complete blockade of Tgf-β receptor signaling by LY-364947 seemed to be predominantly important for prevention of fibrosis in C2C12-SKO cells (Fig 4F and 4G). Interestingly, the inhibition of Tgf-β receptor signaling by LY-364947 also caused lower expression of PGC-1α and AMPK2, but not PPARδ repression (Fig 4H). This may indicates that both PGC-1α and AMPK2 may have different effects on lipid metabolism via Tgf-β receptor signaling as well as via PPARδ nuclear receptor signaling [53, 54].

We also found that C2C12-SKO cells had higher ALP activation compared with C2C12-WT cells (S3E Fig). This may indicate the possibility that osteoblast formation mediated by BMP4 signaling was also involved in myogenic deterioration of C2C12-SKO cells [39]. Taken above data together, these results suggest that SMRT seems to control fibrosis via the multiple transcriptional mediators downstream of Tgf-β family receptor signaling [45, 55].

Although UBC-SKO mice manifested both lower quadricep weight which mimics muscle wasting and higher mRNA levels of some specific Tgf-β pathway signaling isoforms (ex. *Tgf-β2*, *Tgf-β3*, and *β-Catenin*) compared to control, there was no evidence of fibrosis or its markers (Figs 5D, S5A and S5B). Thus, SMRT might have only small effects on the development of fibrosis in adult skeletal muscle rather than during development.

In conclusion, we shed light on SMRT function as a dual suppressor of both fatty acid β-oxidation and Tgf-β receptor signaling which induces fibrosis during the process of C2C12 differentiation and potentially skeletal muscle development. These results support the notion of a potential therapeutic effect of either a SMRT analog or a Tgf-β receptor antagonist for congenital muscle wasting disorders. Herein, we did not elucidate the mechanism by which SMRT deficiency directly or indirectly affected Tgf-β signaling in myocytes. While other studies have reported that a treatment with Tgf-β has direct effect to induce intranuclear SMRT expression in mouse heart and embryonic lung cells the net effect of Tgf-β on the mechanisms of transcriptional regulation by SMRT is still unclear [52, 56]. To clarify the detailed mechanism of SMRT function in the regulation of β-oxidation and Tgf-β receptor signaling further *in vivo*, additional analyses using novel SMRT mouse genetic models will be required.

## Supporting information

**S1 Fig. SMRT protein is strongly expressed in the later phase of differentiation process of C2C12 cells.** (A) The series of time points (day 0–5) of mRNA expression levels of *Smrt* during the differentiation process of C2C12-WT cell were assessed by qPCR (n = 3). (B) The series of time points (day 0–5) of protein expressions of SMRT during the differentiation process of

C2C12-WT cell were assessed by WB (n = 3). (C) Both C2C12-WT and C2C12-SKO2 cells were induced to differentiate in differentiation medium. The expressions of MYH4 in both C2C12-WT and C2C12-SKO2 were assessed by IHC. Myotubes were stained by MYH4 (Green). Nuclei were stained by DAPI (Blue). (D) mRNA expression of *Myh4* gene in both C2C12-WT and C2C12-SKO2 cells was quantified by qPCR (n = 3 per group). (E) Protein expressions of SMRT and MYH4 in both C2C12-WT and C2C12-SKO2 cells were assessed by WB (n = 3 per group). One-way ANOVA was used in (A) and (B) for statistical analyses. The same alphabet on the top of graph bars indicate there is no statistical significance between them. Student t-tests were used for statistical analyses in (C), (D) and (E). Results are shown as the mean±SEM (error bars represent SEM), and the p-value are shown as ***$p < 0.001$, **$p < 0.01$.
(EPS)

**S2 Fig. SMRT suppresses β-oxidation in the pre-differentiation state of C2C12 cells.** (A) The mRNA expressions of β-oxidation-related gene in the pre-differentiation state of both C2C12-WT and C2C12-SKO cells were quantified by qPCR (n = 3 per group). (B) Protein expressions of PPARδ and PGC-1α in the pre-differentiation state of both C2C12-WT and C2C12-SKO cells were assessed by WB (n = 3 per group). (C) Fatty acid oxidation (FAO) rates compared between C2C12-WT and C2C12-SKO cells in the pre-differentiation state were determined (n = 3 per group). (D) The mRNA expression of *Ncor1* in the post-differentiation state in both C2C12-WT and C2C12-SKO cells was quantified by qPCR (n = 3 per group). (E) Protein expression of NCOR1 in the post-differentiation state was assessed by WB (n = 3 per group). Student t-tests were used for statistical analyses in all qPCR and quantification of protein expression. Results are shown as the mean±SEM (error bars represent SEM), and the p-value are shown as ****$p < 0.0001$, **$p < 0.01$, *$p < 0.05$.
(EPS)

**S3 Fig. Proteomics analysis identified the Tgf-β signaling was predominant in C2C12-SKO cells.** (A) Total 397 proteins (44 were significantly up-regulated, and 41 down-regulated) were identified by LC-MS/MS used Triple-TOF®, Sciex 6600 in the comparison between C2C12-SKO and C2C12-WT cells. (B) Lists of top canonical pathways were ranked by IPA proteomics analyses in C2C12-SKO compared with C2C12-WT cells. (C) Causal network analysis identified TGFB1(TGF-β1) as upstream regulator. Rectangular means cytokine, and ellipse means transcriptional regulator. Orange means predicted activation, and blue means inhibition. The color depth indicates intensity of activation. The pointed arrowheads represent activating relationship and a blunt arrowhead inhibitory relationship. The dashed lines indicate virtual relationships composed of the net effect of the pathways between the root regulator and the target molecules. (D) The mRNA expression of Tgf-β families in the pre-differentiation state of both C2C12-WT and C2C12-SKO cells was quantified by qPCR (n = 3 per group). Student t-tests were used for statistical analyses in qPCR. Results are shown as the mean±SEM (error bars represent SEM), and the p-value are shown as ****$p < 0.0001$, ***$p < 0.001$. (E) ALP activities in the pre-differentiation state of both C2C12-WT and C2C12-SKO cells were determined (n = 3 per group).
(EPS)

**S4 Fig. Dual suppression of Tgf-β isoforms ameliorates myotube formation which is limiting to improve fibrosis in the differentiation of C2C12-SKO cells.** (A) C2C12-SKO cells were tested by three groups of si-RNA; si-Control, si-Tgf-β3, and dual knock-down by both si-Tgf-β1 and si-Tgf-β3. They were induced to differentiation in differentiation medium. The expressions of MYH4 were assessed by IHC. Myotubes were stained by MYH4 (Green). Nuclei

were stained by DAPI (Blue). (B) mRNA expressions of three *Tgf-β* isoforms and *Col1a1* in the three groups were quantified by qPCR (n = 3 per group). (C) Protein expressions of MYH4 and COL1A1 in the three groups were assessed by WB (n = 3 per group). For statistical analyses, One-way ANOVA was used in the fusion index panel of (A), (B) and (C). Results are shown as the mean±SEM (error bars represent SEM), and the p-value are shown as \*\*\*\*$p < 0.0001$, \*\*\*$p < 0.001$, \*$p < 0.05$.
(EPS)

**S5 Fig. Global SMRT-null mice showed the loss of skeletal muscle weight.** (A) Quadriceps dissected from both UBC-SKO and control mouse were shown. (B) The quadriceps weights compensated by body weight were measured in both UBC-SKO and control mice (n = 5–6 mice/group).
(EPS)

**S1 Table. Key resources.**
(PDF)

**S2 Table. Comprehensive proteomics analysis of protein expressions in C2C12-WT and C2C12-SKO cells.**
(XLSX)

**S1 Raw images.**
(PDF)

## Acknowledgments

We thank Dr. Ken-ichi Inoue (Center of Regenerative Medicine, Comprehensive Research facilities for Advanced Medical Science, Dokkyo Medical University School of Medicine) for kind assistance on proteomics analysis by IPA software.

## Author Contributions

**Conceptualization:** Hiroaki Shimizu, Hiroyuki Sugimoto, Ronald N. Cohen, Anthony N. Hollenberg.

**Data curation:** Hiroaki Shimizu, Yasuhiro Horibata, Izuki Amano, Megan J. Ritter, Mariko Domae, Hiromi Ando, Hiroyuki Sugimoto.

**Methodology:** Hiroaki Shimizu, Yasuhiro Horibata, Izuki Amano, Megan J. Ritter, Hiroyuki Sugimoto.

**Resources:** Hiroaki Shimizu.

**Supervision:** Hiroyuki Sugimoto, Ronald N. Cohen, Anthony N. Hollenberg.

**Validation:** Hiroyuki Sugimoto, Ronald N. Cohen, Anthony N. Hollenberg.

**Visualization:** Hiroaki Shimizu.

**Writing – original draft:** Hiroaki Shimizu, Ronald N. Cohen, Anthony N. Hollenberg.

**Writing – review & editing:** Hiroaki Shimizu, Ronald N. Cohen, Anthony N. Hollenberg.

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
