## [Decision Letter · Decision Letter 0]

15 Jul 2022

PONE-D-22-11164Nuclear corepressor; SMRT acts as a strong regulator for both β-oxidation and suppressor for fibrosis in the differentiation process of mouse skeletal musclePLOS ONE

Dear Dr. Shimizu,

Thank you for submitting your manuscript to PLOS ONE. After careful consideration, we feel that it has merit but does not fully meet PLOS ONE’s publication criteria as it currently stands. Therefore, we invite you to submit a revised version of the manuscript that addresses the points raised during the review process.

We look forward to receiving your revised manuscript.

Kind regards,

Atsushi Asakura, Ph.D

Academic Editor

PLOS ONE

Journal Requirements:

2. To comply with PLOS ONE submissions requirements, in your Methods section, please provide additional information on the animal research and ensure you have included details on (1) methods of sacrifice, (2) methods of anesthesia and/or analgesia, and (3) efforts to alleviate suffering.

Reviewers' comments:

Reviewer's Responses to Questions

**Comments to the Author**

1. Is the manuscript technically sound, and do the data support the conclusions?

Reviewer #1: Partly

Reviewer #2: Partly

2. Has the statistical analysis been performed appropriately and rigorously? 

Reviewer #1: Yes

Reviewer #2: Yes

3. Have the authors made all data underlying the findings in their manuscript fully available?

Reviewer #1: Yes

Reviewer #2: Yes

4. Is the manuscript presented in an intelligible fashion and written in standard English?

Reviewer #1: No

Reviewer #2: Yes

5. Review Comments to the Author

Reviewer #1: PONE-D-22-11164

Nuclear corepressor; SMRT acts as a strong regulator for both β-oxidation and suppressor for fibrosis in the differentiation process of mouse skeletal muscle.

In this paper, the authors found that loss of SMRT in C2C12 cells reduces muscle differentiation potential and increases fibrosis-related factors through enhanced TGFb signaling. Indeed, they have shown that suppression of TGFb expression by siRNA and suppression of TGF signaling by LY364947 can restore the myogenic differentiation potential of SMRT-deficient C2C12 cells as well as reduce the expression of fibrosis-associated factors. However, these in vitro findings could not necessarily be reproduced in vivo, and further validation is expected.

There are no particular concerns with the experiments themselves, and the individual results obtained are clear. However, the poor writing and the poor logic of the paper make it extremely difficult to understand.

1. SKO-C2C12 in Fig. 1A shows no myotube formation, while SKO-C2C12 in the control group shown in Fig. 4A and 4D does show myotube formation, albeit weak. What is the reason for this difference?

2. The authors used GAPDH and POL-2 as loading controls in western blot. How is it used differently? Is it the molecular weight of the factor to be observed?

3. In the graph quantifying Western blot, the expression "fold change" is used on the vertical axis, but the value shown is relative to the loading control. Therefore, the expression "fold change" is inappropriate, and it would be better to use "relative expression”.

4. It is difficult to understand the logic in writing. For example, in Fig. 2, there are cases where gene expression does not necessarily correlate with actual protein levels (PGC-1a and HDAC3). Nevertheless, in Fig. 3, to verify the results of Proteomics analysis, the changes in gene expression were examined by qPCR, THEN, the protein levels were further analyzed by western blot. It is unclear why the logic behind the verification of gene expression by qPCR is so daring here.

5. What does the “FGF2 fusion protein” in Fig. 3C represent?

6. The western blot of BMP4 shows a comparison of protein levels of precursor form, but TGFb family factors are only active when the pro-domain is cleaved after forming a dimer. Is not it necessary to look at the active form?

7. Similarly, in Fig. 4, only the gene expression of TGFb is shown, but the actual amount of active TGFb is correlated with the gene expression of TGFb?

8. I strongly recommend the authors get proofread on the whole English text.

Reviewer #2: This manuscript written by Dr. Shimizu and colleagues aimed to clarify the functions of SMRT during myogenic cell differentiation, and found that SMRT could regulate myogenic differentiation by preventing fibrosis via TGF-b signaling.

This study is very interesting, and well designed.

However, one of the critical concerns in this study was that even though SKO myoblasts did not form myotubes (Figure 1), the group compared gene and protein expression in post-differentiation myotubes derived from WT and SMRT-deficient (SKO) cells. The reviewer wonders whether the alterations in gene and protein expression observed in this study reflected differences between WT and SMRT-deficiency or between myoblasts and myotubes. The group should carefully examine and discuss this point. Another concern was that it was unclear whether SMRT deficiency directly or indirectly affected beta-oxidation or TGFb signaling activation. The reviewer raised some concerns that need to be addressed in order to strengthen the conclusions drawn by the authors.

In Figure 1A, though the group established two clones of SMRT-deficient C2C12 cells, the authors only showed data for one clone. Please explain If there is a reason for this.

In figure 2A and B, the authors have shown the data about the beta-oxidation related gene and protein expressions using post-differentiated myotubes, and have found that the increase of these expressions in SKO-cells. However, as shown in figure 1, since SKO did not form myotubes after differentiation, these alterations observed in gene and protein expression between WT and SKO might not reflect the differences in myotube from WT and SKO, but in myotubes and myoblasts. Please discuss these points

In figure 2, the results have shown the increase of beta-oxidation related gene and protein expressions, the authors should assess the abilities of fatty-acid oxidation rate in SKO-myoblasts or myotubes.

In figure 3 and 4, SMRT-deficiency impaired myotube formation with the increase of TGFb expressions, and inhibition of TGFb signaling restored the potential for myogenic differentiation. However, it is not clear why SMRT-deficiency induced TGFb signaling. Please discuss this point. In addition, to confirm whether TGF-b signaling affected myogenic cell differentiation, please evaluate the expression of p-SMAD2 and p-SMAD3 after treatment with siTGFb1 or LY-364947 in SKO-myoblasts or myotubes.

In figure 4, suppression of TGF-b signaling by siTGFb1 and LY-364947 treatment could improve myogenic cell differentiation. However, siTGFb1 treatment in SKO myoblasts increased Col1A expression while this expression was decreased in LY-364947 treated SKO-myoblasts (Figure 4C and F). Please explain this difference. Additionally, please evaluate the expression of beta-oxidation related genes including PPARg, PGC-1a, and Ampk2 to see whether suppressing TGF-b signaling decreases lipid metabolism.

In Figure 5, SMRT-KO muscles have shown the decrease in expression of beta-oxidation-related protein, which is inconsistent with the results observed in C2C12 SKO-myoblasts or myotubes. Please explain these discrepancies.

In the section of Methods, the group analyzed two different protein expressions (POL-2 and GAPDH) as internal controls. Please explain why the authors applied them separately. In figure 3C, COL1A1 expression was normalized by GAPDH, while was normalized by POL-2 in Figure 4C.

6. PLOS authors have the option to publish the peer review history of their article (what does this mean?). If published, this will include your full peer review and any attached files.

Reviewer #1: No

Reviewer #2: No

---

## [Author Response · Author response to Decision Letter 0]

13 Oct 2022

To the Editors:

We thank the academic editor and reviewers for their comments on our manuscript. Below are our point by point responses to each comment. We hope that our manuscript is now acceptable for publication.

In the “Revised Manuscript with Track Changes”, newly inserted words were highlighted by yellow, and deleted words were shown by red with strikethorough.

Journal Requirements:

Thank you for the suggestion. We read through them, and we think that now our manuscript meets the journal requirements.

2. To comply with PLOS ONE submissions requirements, in your Methods section, please provide additional information on the animal research and ensure you have included details on (1) methods of sacrifice, (2) methods of anesthesia and/or analgesia, and (3) efforts to alleviate suffering.

To comply submission requirements, we provided additional animal research information in the method section; page4, line127.

Thank you, we provided all original blot images in the Supporting information files named as “S1_raw images”.

Reviewing all raw blot data, we made three corrections in the original figures and manuscript.

1: The exact molecular weight (MW) of collagen type 1 (COL1A1) protein is 250kDa. All the MWs of COL1A1 in Fig3C, 4C, 4F, and S4 FigC were corrected. Also the picture of COL1A1 in Fig3C was corrected.

2: We deleted two images of ALK5 (Fig3E) and MYH (Fig5A).

3: Since the original blot of BMP4 in Fig3C was different from others (COL1A1 and FGF2 isoform) in the same section, we separately displayed the blots of both BMP4 and its corresponding GAPDH.

We would appreciate if you could consider these corrections for reviewing.

Thank you for the suggestion, in the cover letter, we mentioned that all original blot images were provided in the Supporting information files.

Reviewers' comments:

5. Review Comments to the Author

Reviewer #1: PONE-D-22-11164

Nuclear corepressor; SMRT acts as a strong regulator for both β-oxidation and suppressor for fibrosis in the differentiation process of mouse skeletal muscle.

In this paper, the authors found that loss of SMRT in C2C12 cells reduces muscle differentiation potential and increases fibrosis-related factors through enhanced TGFb signaling. Indeed, they have shown that suppression of TGFb expression by siRNA and suppression of TGF signaling by LY364947 can restore the myogenic differentiation potential of SMRT-deficient C2C12 cells as well as reduce the expression of fibrosis-associated factors. However, these in vitro findings could not necessarily be reproduced in vivo, and further validation is expected.

There are no particular concerns with the experiments themselves, and the individual results obtained are clear. However, the poor writing and the poor logic of the paper make it extremely difficult to understand.

Thank you for the suggestion, we hope that the revised manuscript is now well-proofread in English and acceptable for publication.

1. SKO-C2C12 in Fig. 1A shows no myotube formation, while SKO-C2C12 in the control group shown in Fig. 4A and 4D does show myotube formation, albeit weak. What is the reason for this difference?

Thank you for the comment. Although the myotubes seem to be poor in the shown field of SKO-C2C12 in Fig.1A, the cells do have weak myotube formation. As mentioned in the method section (page6; line176-179), we chose five fields randomly from same picture to calculate the fusion index. We think this can offset the small difference of myotube formation observed in each taken field.

2. The authors used GAPDH and POL-2 as loading controls in western blot. How is it used differently? Is it the molecular weight of the factor to be observed?

Thank you for the comment, this is just for a technical reason. For the assessment of protein amount, we compared each different protein band with the corresponding loading control band which has a close molecular weight on the same blot membrane. Because of this reason, we assessed the expression of COL1A1 in figure 3C was normalized by GAPDH as we also probed for the FGF2 isoform which has a lower molecular weight. In contrast, we used POL-2 in Figure 4C as lower molecular weight proteins were not included. To avoid confusion, we also added further explanation in the method section of revised manuscript (page6; line189-198).

3. In the graph quantifying Western blot, the expression "fold change" is used on the vertical axis, but the value shown is relative to the loading control. Therefore, the expression "fold change" is inappropriate, and it would be better to use "relative expression”.

We really agree with your comment. All the expressions of “fold change” were changed to “relative expression” in all figures. Also, we re-phrased the explanation for calculation of “relative expression” in the method section in page6; line189-198.

4. It is difficult to understand the logic in writing. For example, in Fig. 2, there are cases where gene expression does not necessarily correlate with actual protein levels (PGC-1a and HDAC3). Nevertheless, in Fig. 3, to verify the results of Proteomics analysis, the changes in gene expression were examined by qPCR, THEN, the protein levels were further analyzed by western blot. It is unclear why the logic behind the verification of gene expression by qPCR is so daring here.

Thank you for the comment. The reason why we showed qPCR results of TGF-beta isoforms and receptor targets in Fig 3 at first is that we would like to confirm SMRT deletion did affect transcriptional derepression of TGF-beta signaling. To understand this logic clearly, we rephrased the explanation in results section (page12; line393-394, 399-400, 416-418, and page13; line430-431).

5. What does the “FGF2 fusion protein” in Fig. 3C represent?

Thank you for the comment. We looked at the 34kDa band which is one of the FGF2 isoforms in Fig 3C. To make this clearly, we rephrased this word as “FGF2 isoform”.

6. The western blot of BMP4 shows a comparison of protein levels of precursor form, but TGFb family factors are only active when the pro-domain is cleaved after forming a dimer. Is not it necessary to look at the active form?

We agree this comment, thank you. We think that the measurement of alkaline phosphatase (ALP) activity in C2C12 myoblasts would be a good indicator for BMP4 signal activation, because Nojima J, et al. (ref.#39) reported that osteoblast formation accompanied with high ALP activity was strongly induced by BMP4 treatment in C2C12 myoblasts. We performed measurement of ALP activity in myoblasts compared with C2C12-WT and C2C12-SKO cells. We added explanations in the method section (page7; line214-218, page8; line260), result (page12; line405-415), conclusion (page19; line642-647), and supplemental figure legend (page26; line894-896).

7. Similarly, in Fig. 4, only the gene expression of TGFb is shown, but the actual amount of active TGFb is correlated with the gene expression of TGFb?. 

Thank you for the comment. To assess the net effect of activated TGF-beta signaling, we investigated the protein expression of both p-SMAD2 and p-SMAD3 compared between two pairs of C2C12-SKO cells treated by vehicle and TGF-beta receptor inhibitor; LY-364947. The result is shown as newly inserted Fig 4G, and explanations were added in the revised manuscript (page14; line490-496, page15; line523-524, and page18; line 631-637). 

8. I strongly recommend the authors get proofread on the whole English text.

Thank you for the recommendation, we hope that the revised manuscript is now well-proofread in English.

Reviewer #2: This manuscript written by Dr. Shimizu and colleagues aimed to clarify the functions of SMRT during myogenic cell differentiation, and found that SMRT could regulate myogenic differentiation by preventing fibrosis via TGF-b signaling.

This study is very interesting, and well designed.

However, one of the critical concerns in this study was that even though SKO myoblasts did not form myotubes (Figure 1), the group compared gene and protein expression in post-differentiation myotubes derived from WT and SMRT-deficient (SKO) cells. The reviewer wonders whether the alterations in gene and protein expression observed in this study reflected differences between WT and SMRT-deficiency or between myoblasts and myotubes. The group should carefully examine and discuss this point.

Thank you for the comment. To distinguish two cell states between pre-differentiation myoblasts and post-differentiation myotubes, we compared the expression analyses of WT and SKO cells in both cell states. The results of post-differentiation myotubes are shown in Figure 2A, B, and the results of pre-differentiation myoblasts were shown in Supplement 2 figure A, B.

To delineate this more clearly, we added the titles of “post-differentiation state (myotube)” on each panel of Figure 2 and Figure 3, and also added the titles of “pre-differentiation state (myoblast)” on each panel of the Supplement2 and Supplement3 figure.

Another concern was that it was unclear whether SMRT deficiency directly or indirectly affected beta-oxidation or TGFb signaling activation. The reviewer raised some concerns that need to be addressed in order to strengthen the conclusions drawn by the authors.

Thank you for the comment. We could not conclude whether SMRT deficiency directly or indirectly affected beta-oxidation and TGF-beta signaling in the myoblasts in this study. Some recent studies done by others reported that a treatment with TGF-beta has a direct effect to induce internuclear SMRT expression in the mouse heart and embryonic lung cell (ref.#52,56). However, the net effect of TGF-beta on the mechanisms of transcriptional regulation by SMRT is still unclear. We add explanations and next perspectives for this in the section of discussion (page19; line658-663). 

In Figure 1A, though the group established two clones of SMRT-deficient C2C12 cells, the authors only showed data for one clone. Please explain If there is a reason for this.

Thank you. As shown in both Fig.1A-C and Supplement 1 Fig.C-E, we confirmed that two clones (C2C12-SKO and C2C12-SKO2) had the same phenotype which decreases myotube formation and MYH4 expression at the beginning of analyses. Then we focused on one of two clones (C2C12-SKO) and performed further analyses to investigate the cause of myotube formation impairment (page9; line284-294, 313-314). To understand this logic clearer, we also added one sentence at the beginning of second paragraph in the results section (page9; line 313-314).

In figure 2A and B, the authors have shown the data about the beta-oxidation related gene and protein expressions using post-differentiated myotubes, and have found that the increase of these expressions in SKO-cells. However, as shown in figure 1, since SKO did not form myotubes after differentiation, these alterations observed in gene and protein expression between WT and SKO might not reflect the differences in myotube from WT and SKO, but in myotubes and myoblasts. Please discuss these points

Thank you. We agree with your comments. As shown in Supplement 2 Fig B and C, both beta-oxidation genes and FAO activity were up-regulated in SKO-myoblast cells even before myotube differentiation, so we think the alterations of beta-oxidation genes reflect the differences between WT-myoblasts and SKO-myoblasts. We put additional explanation of this in page10; line324-329.

In figure 2, the results have shown the increase of beta-oxidation related gene and protein expressions, the authors should assess the abilities of fatty-acid oxidation rate in SKO-myoblasts or myotubes. 

Thank you. We measured the FAO activity in SKO-myoblast by fluorescence detection assay and added this in the methods section (page6-7; line200-212), results (page10; line324-329), and figure legend (page25; line870-872).

In figure 3 and 4, SMRT-deficiency impaired myotube formation with the increase of TGFb expressions, and inhibition of TGFb signaling restored the potential for myogenic differentiation. However, it is not clear why SMRT-deficiency induced TGFb signaling. Please discuss this point.

Thank you for your comment. We inserted additional explanations and expressed our opinion for your question in the section of discussion (page19; line658-663).

In addition, to confirm whether TGF-b signaling affected myogenic cell differentiation, please evaluate the expression of p-SMAD2 and p-SMAD3 after treatment with siTGFb1 or LY-364947 in SKO-myoblasts or myotubes.

We evaluated the expression of both p-SMAD2 and p-SMAD3 after treatment with LY364947 in SKO-myoblasts. Fig.4G was newly assigned as a figure number for this blot, and the explanation was added in the manuscript (page14; line493-496, page15; line523-524, page18; line633-637).

In figure 4, suppression of TGF-b signaling by siTGFb1 and LY-364947 treatment could improve myogenic cell differentiation. However, siTGFb1 treatment in SKO myoblasts increased Col1A expression while this expression was decreased in LY-364947 treated SKO-myoblasts (Figure 4C and F). Please explain this difference. 

Thank you for the comment. Based on the results of siTGFb treatment in Fig. 4A-C and Supplement 4 Fig. A-C, we think that the inhibition of one of three TGFb isoforms can improve myotube formation, but not fibrosis. On the other hand, the interruption of the TGFb type I receptor signal by LY-364947 seemed to prevent fibrosis as well as myotube differentiation. To confirm this, we additionally evaluated the expression of p-SMAD2 and p-SMAD3 after treatment with LY364947 in SKO-myoblasts (Fig 4G). We also put an additional explanation in the section of discussion (page18; line633-637).

Additionally, please evaluate the expression of beta-oxidation related genes including PPARg, PGC-1a, and Ampk2 to see whether suppressing TGF-b signaling decreases lipid metabolism.

To evaluate the expression of beta-oxidation related gene after treatment with LY364947 in SKO-myoblasts, Fig.4H was newly assigned as a figure number for these blots, and explanations were added in the manuscript in page15; line497-501, 504-505, 525-526, and page18-19; line631-641.

In Figure 5, SMRT-KO muscles have shown the decrease in expression of beta-oxidation-related protein, which is inconsistent with the results observed in C2C12 SKO-myoblasts or myotubes. Please explain these discrepancies.

Thank you for the comment. We put the additional discussion in the conclusion section of revised manuscript on page17-18; line602-607.

In the section of Methods, the group analyzed two different protein expressions (POL-2 and GAPDH) as internal controls. Please explain why the authors applied them separately. In figure 3C, COL1A1 expression was normalized by GAPDH, while was normalized by POL-2 in Figure 4C.

Thank you for the suggestion. We added further explanations for this comment in the method section of revised manuscript on the page6; line195-198.

---

## [Editor Report · Decision Letter 1]

4 Nov 2022

Nuclear corepressor SMRT acts as a strong regulator of both β-oxidation and suppressor of fibrosis in the differentiation process of mouse skeletal muscle cells

PONE-D-22-11164R1

Dear Dr. Shimizu,

We’re pleased to inform you that your manuscript has been judged scientifically suitable for publication and will be formally accepted for publication once it meets all outstanding technical requirements.

Kind regards,

Atsushi Asakura, Ph.D

Academic Editor

PLOS ONE
---

## [Editor Report · Acceptance letter]

10 Nov 2022

PONE-D-22-11164R1 

Nuclear corepressor SMRT acts as a strong regulator of both β-oxidation and suppressor of fibrosis in the differentiation process of mouse skeletal muscle cells 

Dear Dr. Shimizu:

I'm pleased to inform you that your manuscript has been deemed suitable for publication in PLOS ONE. Congratulations! Your manuscript is now with our production department. 

Kind regards, 

on behalf of

Dr. Atsushi Asakura 

Academic Editor

PLOS ONE